# Molecular Diet Analysis of Asian Clams for Supplementary Biodiversity Monitoring: A Case Study of Nakdong River Estuary

**DOI:** 10.3390/biology12091245

**Published:** 2023-09-16

**Authors:** Kanghui Kim, Gea-Jae Joo, Kwang-Seuk Jeong, Jeong-Soo Gim, Yerim Lee, Donghyun Hong, Hyunbin Jo

**Affiliations:** 1Department of Integrated Biological Science, Pusan National University, Busan 46241, Republic of Korea; kgh780@naver.com (K.K.); gjjoo@pusan.ac.kr (G.-J.J.); revenant1@naver.com (J.-S.G.); yr2884@naver.com (Y.L.); hdh1201@pusan.ac.kr (D.H.); 2Department of Nursing Science, Busan Health University, Busan 49318, Republic of Korea; kjeong@bhug.ac.kr

**Keywords:** *Corbicula fluminea*, eDNA metabarcoding, conventional field survey, next-generation sequencing, 18S V9

## Abstract

**Simple Summary:**

Filter feeders can retain environmental DNA (eDNA) within their bodies, making them potential eDNA samplers. In this study, eDNA from the gut contents of Asian clams (*Corbicula fluminea*) was used to identify biodiversity in estuarine ecosystems. Various organisms, such as fish, copepods, and green algae, were detected, representing a wide range of habitats. Of the 20 families detected (except for Fungi and terrestrial taxa), 8 families were also documented in the conventional field survey, enabling the identification of an operational taxonomic unit (OTU) of migratory fish that are challenging to observe directly. These results support the potential application of *C. fluminea* as a supplementary tool for investigating the biodiversity of aquatic ecosystems.

**Abstract:**

Environmental DNA (eDNA) extracted from the gut contents of filter feeders can be used to identify biodiversity in aquatic ecosystems. In this study, we used eDNA from the gut contents of the Asian clam *Corbicula fluminea* to examine biodiversity within estuarine ecosystem. Field sampling was conducted at three points in the Nakdong River Estuary, which is characterised by closed estuarine features resulting from the presence of an estuarine barrage. The collected *C. fluminea* samples were dissected to separate the gut contents, and the extracted eDNA was amplified using 18S V9 primer targeting all eukaryote-derived DNA. The amplified DNA was sequenced using a next-generation sequencing (NGS) technique, and a BLASTn search was performed based on the National Centre for Biotechnology Information (NCBI) database for taxa identification. We obtained 23 unique operational taxonomic units (OTUs), including fish (approximately 8.70%), copepods (approximately 17.39%), and green algae (approximately 21.74%), representing a wide range of habitats. Furthermore, 8 out of the 20 families were identified through comparisons with reference data from conventional field surveys, and the OTUs of elusive migratory fish were detected. The results support the application of *C. fluminea* as an eDNA sampler for supplementary biodiversity monitoring.

## 1. Introduction

Estuaries are highly complex and dynamic ecosystems that provide multifaceted habitats, such as tidal mudflats, sandbars, marshes, and transition zones, for numerous organisms [1,2,3]. However, estuaries are often subjected to serious threats from anthropogenic impacts, including overexploitation, reclamation, pollution, and barrage construction, leading to rapid declines in habitats and biodiversity [4,5]. Therefore, monitoring and detecting critical changes in estuarine ecosystems are important [6]. Conventional aquatic species monitoring methods that capture or rely on direct detection are time-consuming and labour-intensive [7,8]. Morphological identification using the naked eye or a microscope is skill-dependent and can lead to misidentification [9]. Furthermore, the dynamic environment and high biodiversity of estuarine ecosystems complicate monitoring [10].

Environmental DNA (eDNA) metabarcoding is a promising alternative to traditional monitoring methods [11]. This molecular technique enables the identification of an entire community from a single environmental sample (e.g., water, soil, air, faeces, or gut contents) without directly observing or capturing organisms. Through polymerase chain reaction (PCR) using universal or group specific primers and next-generation sequencing (NGS) techniques, researchers can identify the presence of various organisms, including rare, elusive, or endangered species [11,12]. This method provides a non-invasive and efficient way to assess biodiversity in different habitats [13]. Consequently, studies employing eDNA metabarcoding for biodiversity assessment have attracted significant attention in recent years and have explored various potential eDNA sources such as biofilms, faeces, or gut contents [14,15,16,17,18,19].

Filter feeders, which filter water and ingest organic particles, can accumulate eDNA within their bodies without an artificial filtering process and can be used to identify biodiversity in aquatic ecosystems [18,19,20]. Their unique and effective feeding habits make them potential eDNA samplers [18,21]. In particular, bivalves such as clams and mussels are widely distributed across aquatic ecosystems, including lakes, rivers, and estuaries, demonstrating broader applicability than other filter feeders [22]. However, there are few studies that have extracted eDNA from bivalves, and their potential has rarely been investigated, especially in specific ecosystems such as closed estuaries.

In this study, we conducted a first-of-its-kind investigation applying *Corbicula fluminea* to eDNA metabarcoding for biodiversity monitoring in a closed estuary, using the Nakdong River Estuary as a case study. This estuary exemplifies an artificially regulated ecosystem owing to the presence of an estuarine barrage, which was constructed in 1987 and reopened in 2020 to restore a brackish ecosystem [23]. The bivalve *C. fluminea* is widely distributed in the Nakdong River Estuary. We hypothesised that eDNA analysis of *C. fluminea* gut contents could reveal the biodiversity of the Nakdong River Estuary.

To test this hypothesis, we analysed eDNA from the gut contents of *C. fluminea* collected from the Nakdong River Estuary. DNA metabarcoding using 18S V9 primers was employed to explore comprehensive biodiversity of the ecosystem. We investigated the composition of detected taxa by analysing parameters such as the number of OTUs, frequency of occurrence (FOO), and relative read abundance (RRA). We subsequently reviewed the applicability of *C. fluminea* as an eDNA sampler through comparison with the actual biological monitoring report conducted at the Nakdong River Estuary.

## 2. Materials and Methods

### 2.1. Study Site

The Nakdong River is the second largest river system in South Korea and maintains a well-developed estuarine system (35°05′ N, 128°55′ E). It is also recognised as an important biodiversity conservation area, including winter bird habitats and stopover sites on the East Asia–Australasian Flyway [24]. Another important feature of the estuary is its flood control activities, primarily through an estuarine barrage built in 1987 which divides brackish areas into distinct freshwater and saline zones. This division is believed to influence biodiversity changes [23], necessitating thorough and efficient monitoring methods to address growing concerns regarding regional biodiversity protection. In the present study, we selected three points within the brackish area with different salinity levels (Figure 1). Points 1, 2, and 3 were located approximately 2.0 km, 2.7 km, and 3.9 km from the estuarine barrage, respectively. *C. fluminea* is widely distributed in this area, as they have been continuously released for fishery resources by the Busan Marine Fisheries Resources Research Institute [25].

### 2.2. Water Quality Survey

A water quality survey was conducted in September 2021 concurrently with the sampling of *C. fluminea*. Water samples were collected from the surface layer (at a depth of approximately 0.5 m) using a 10 L polypropylene bucket. Dissolved oxygen (DO, mg L^−1^, %) and water temperature (°C) were measured using a YSI 550A dissolved oxygen instrument (YSI, Yellow Springs, OH, USA). The pH levels were determined using a YSI Model 60 handheld pH–temperature system (YSI, Yellow Springs, OH, USA). Electrical conductivity (μS cm^−1^) and salinity (ppt) were assessed using a YSI Pro30 conductivity meter (YSI, Yellow Springs, OH, USA). Following the field survey, water samples were transported to the laboratory in refrigerated storage for turbidity and alkalinity analyses. Turbidity (NTU) was measured using an APERA TN500 portable white light turbidity meter (APERA, Columbus, OH, USA), and alkalinity (mg L^−1^) was determined using the neutralisation method in accordance with standard procedures [26].

### 2.3. C. fluminea Sampling and Pretreatment

*Corbicula fluminea* samples were collected using a fishing dredge 123 cm wide and 22 cm high (Figure A1). The dredge net was made of polyethylene and was 320 cm long with an 11 × 11 mm mesh size. The collected *C. fluminea* samples were placed in separate polyethylene bags at each point and transported to the laboratory for refrigerated storage. The samples were stored at −80 °C until further analysis.

From the collected *C. fluminea* samples, 10 individuals per point were randomly selected (totalling 30 mature individuals; shell length (cm), 2.0–3.0; shell height (cm), 1.9–2.4; shell width (cm), 1.1–1.5; and total weight (g), 2.990–6.051) [27]. The gut was eviscerated and dissected to obtain its contents. During dissection, scalpels, tweezers, and scissors were flame-sterilised between samples to minimise contamination. The extracted gut contents were placed in 1.5 mL microtubes separately (*n* = 30) and stored at −20 °C until further analysis.

### 2.4. DNA Extraction and Amplification

The gut contents from the *C. fluminea* samples were homogenised using sterilised homogeniser pestle, and genomic DNA was extracted using DNeasy Blood & Tissue Kit (Qiagen, Hilden, Germany), according to the manufacturer’s instructions. The extracted DNA samples were stored at −20 °C.

Two consecutive PCR steps were performed for the next-generation sequencing (NGS) process. Throughout the entire PCR process, both negative and positive controls were employed. The first PCR was performed using primer sets that amplify the V9 regions of 18S rRNA (18S V9 primer) targeting universal eukaryotes. The forward primer sequence is 5′-TCGTCGGCAGCGTCAGATGTGTATAAGAGACAGCCCTGCCHTTTGTACACAC-3′, and the reverse is 5′-GTCTCGTGGGCTCGGAGATGTGTATAAGAGACAGCCTTCYGCAGGTTCACCTAC-3′. We used AccuPower HotStart PCR PreMix (Bioneer, Deajeon, Republic of Korea), and the volume of the PCR reaction solution was 20 µL^−1^ (DNA template 1 µL, forward primer 1 µL, reverse primer 1 µL, and distilled water 17 µL). The PCR conditions consist of 1 cycle of initial denaturation (94 °C, 10 min) and 35 cycles of denaturation (94 °C, 1 min), annealing (50 °C, 1.5 min), extension (72 °C, 1 min) and 1 cycle of final extension (72 °C, 10 min). After first PCR, we confirmed the size of the products through 1.5% agarose gel electrophoresis and stored them at −20 °C.

The second PCR was performed using KAPA HiFi HotStart ReadyMix (KAPA Biosystems, Wilmington, MA, USA) and Nextera XT Index Kit v2 (Illumina, San Diego, CA, USA). The volume of the PCR reaction solution was 25 µL^−1^ (first PCR product 2.5 µL, Forward index 2.5 µL, Reverse index 2.5 µL, KAPA mix 12.5 µL, distilled water 5 µL). The PCR conditions consist of 1 cycle of initial denaturation (95 °C, 3 min) and 10 cycles of denaturation (95 °C, 30 s), annealing (55 °C, 30 s), extension (72 °C, 30 s), and 1 cycle of final extension (72 °C, 5 min). Subsequently, we confirmed the size of the products through 1.5% agarose gel electrophoresis and stored them at −20 °C.

The PCR products were purified by a beads clean-up process using AMPure XP Reagent (Beckman Coulter, Indianapolis, IN, USA), and then pooled in equal concentration (10 nM) using a DeNovix QFX Fluorometer and a DeNovix dsDNA Ultra High Sensitivity Assay (DeNovix, Wilmington, DE, USA) according to the manufacturer’s protocol. The generated library was stored at −20 °C until DNA sequencing.

### 2.5. DNA Sequence Analysis

We sequenced library samples using NGS and performed taxonomic identification. The library was sequenced on an Illumina iSeq platform (Illumina, San Diego, CA, USA), and data processing was performed using USEARCH (v11.0.667) [28]. Demultiplexed raw sequences (FASTQ files) were merged into one sequence, allowing a maximum of ten mismatches. Merged reads with expected errors > 1.0 were discarded after quality filtering. The remaining sequences were dereplicated and clustered into OTUs at a 97% OTU cut-off value, removing chimeric and singleton sequences.

The resulting OTU sequences were searched in the National Centre for Biotechnology Information (NCBI) database (Release 255.0; July 2023) using BLASTn [29]. A list of the 200 taxa for each OTU with the highest max score was obtained. The OTUs were provisionally identified based on identity percentages, with OTUs exhibiting an identity of 97% or higher assigned at the species level, whereas the remaining (90–97%) were assigned at the genus or family level. Subsequently, we referenced the biodiversity database of South Korea (National Institute of Biological Resources) [30] to ascertain the presence of the taxa within the nation’s territory. In cases wherein the taxa were not confirmed in the reference, we considered either the higher taxonomic ranks of the taxa or second-score taxa. The OTUs identified as *C. fluminea* were considered ‘self-DNA’ and excluded from the subsequent process. The obtained sequences were deposited in the NCBI repository under accession number SAMN35796656.

Next, we categorised the identified OTUs according to taxonomic classifications and examined habitat environments, based on the World Register of Marine Species (WoRMS) database (Table A1) [31]. For *Hygrobates* sp., as it was not found in the WoRMs database, we used the NCBI taxonomy browser [32] and additional references instead [33].

### 2.6. Statistical Analysis and Reference Data

To indicate the overall taxa composition detected in the gut contents of *C. fluminea*, the number of OTUs was presented based on the taxonomic categories, and two dietary metrics were used, FOO and RRA [34]. The FOO refers to the proportion of samples in which a particular taxon was detected. FOO of a taxon indicates how frequently it appears across all the samples in a study. A high FOO for a taxon suggests that it is widespread and common in the study area, whereas a low FOO indicates that it is less prevalent. The RRA represents the proportion of sequence reads that belonging to a specific taxon relative to the total number of reads obtained during the sequencing process. A high RRA for a taxon indicates that it is abundant and represented by a large number of reads, whereas a low RRA suggests that it is less abundant in the sample.

We examined the differences in the detected taxa composition between three sampling points using a Permutational Multivariate Analysis of Variance (PERMANOVA) performed with 999 random permutations. The analysis was conducted in PRIMER v7 (PRIMER-e, Auckland, New Zealand) on the basis of a Bray–Curtis dissimilarity matrix and using log(x + 1) transformed RRA data.

To assess the correspondence between taxa identified from the gut contents of *C. fluminea* and the actual biodiversity of the Nakdong River Estuary, we referred to a monitoring report published by the Korea Water Resources Corporation (K-water) [35]. The conventional field survey was conducted throughout 2021 (April, June, July, August, October, and November) in both the upstream and downstream areas of the estuarine barrage, encompassing a broader spatiotemporal range, including the timing and locations of our *C. fluminea* sampling (September 2021, downstream of the barrage). Reference data included the survey results for phytoplankton, zooplankton, benthic invertebrate, shellfish, fish, and vegetation. In the present study, we considered the data of phytoplankton, zooplankton and fish, aiming to ascertain whether the taxa detected in present study were also documented in the conventional field survey. Additionally, we referred only to the results of formal surveys conducted on a regular basis, and other additional data were subsidiarily considered.

## 3. Results

### 3.1. Water Parameters

The water parameters did not differ considerably among the sampling points, except for salinity and electrical conductivity (Table 1). Salinity values ranged from 10.0 to 14.2 ppt, indicating a brackish area. They increased in the following order: Points 1, 3, and 2, regardless of the distance from the estuarine barrage. Electrical conductivity exhibited a trend similar to salinity, as it was also affected by dissolved salts.

### 3.2. eDNA from the Gut Contents of C. fluminea

The DNA metabarcoding analysis generated 17,272 paired-end reads from 30 samples. After quality filtering, 16,980 (98.3%) sequences were obtained, comprising 53 unique OTUs. Through a series of identification processes, 23 eukaryotic taxa were identified (belonged to 22 genera, 22 families, 17 orders, 15 classes, 12 phyla, and 5 kingdoms; Table 2) in 28 samples. We assigned 3 OTUs at the species level, 18 OTUs at the genus level, and 2 OTUs at the family level.

The OTUs identified in the *C. fluminea* gut contents represented taxonomically diverse organisms (Figure 2A). Animalia accounted for the highest percentage (nine OTUs), followed by Chromista (six OTUs), Plantae (five OTUs), Fungi (two OTUs), and Protozoa (one OTU). The OTUs in the kingdom Animalia included three phyla, with copepods (Arthropoda) and fish (Chordata) accounting for 17.39% (4 OTUs) and 8.70% (2 OTUs) of the total 23 OTUs, respectively. The OTUs in the kingdom Chromista covered five phyla, with diatom (Bacillariophyta) representing the highest proportion (8.70%, 2 OTUs). All Plantae OTUs belonged to the phylum Chlorophyta (green algae).

FOO and RRA demonstrated the detection frequency and relative abundance of each taxon recovered from *C. flumina* gut contents (Figure 2B,C). *Clausidium* sp. (copepods) exhibited the highest proportion in both FOO and RRA, being present in the greatest number of samples (93.33%) and displaying the most abundant DNA reads (66.78%). Subsequently, *Strombidium* sp. (ciliates) appeared in 86.67% of the samples, accounting for 9.67% of the DNA reads, whereas *Hygrobates* sp. (water mites) appeared in 40.00% of the samples, accounting 4.05% of the DNA reads. Additionally, *Oncorhynchus* sp. (salmon and trout; 33.33% FOO, 2.23% RRA), *Mychonastes* sp.1 (green algae; 30.00% FOO, 1.06% RRA), *Paragonimus* sp. (lung fluke; 23.33% FOO, 11.47% RRA), and *Mytilicola* sp. (parasitic copepods; 10.00% FOO, 1.63% RRA) displayed RRA values >1%. 

Furthermore, the identified OTUs covered various habitat environments, including marine, brackish, freshwater, and terrestrial environments (Table A1). They comprised freshwater green algae (*Desmodesmus communis*), marine copepods (*Clausidium* sp., *Anthessius* sp.), and even terrestrial insects (*Liposcelis* sp.).

### 3.3. Comparison between Sampling Points

In the comparison of number of OTUs, RRA, and FOO, no marked difference was observed between the three sampling points (10 samples per sampling point). The Points 1, 2 and 3 included 19 OTUs (11 phyla, 4 kingdoms), 13 OTUs (8 phyla, 5 kingdoms), 16 OTUs (10 phyla, 5 kingdoms) of the total 23 OTUs, respectively (Figure 3A). At all sampling points, the kingdom Animaila accounted for ≥90% FOO and ≥85% RRA, whereas Chromista accounted for ≥80% FOO and ≥7% RRA (Figure 3B,C). Based on PERMANOVA, the three sampling points did not exhibit significant differences in their RRA data of total 23 taxa (Table A2; *p* > 0.05), suggesting that the composition of taxa detected from the *C. fluminea* gut contents was not different across the sampling points.

Additionally, we cross-checked the appearance of OTUs among the sampling points (Figure 4, the red circles). Of the total 23 detected taxa, 10 OTUs (10 families) appeared at all three sampling points, 5 OTUs (5 families) appeared at two points, and 8 OTUs (7 families) appeared at only one point. There were no remarkable taxonomical trends based on the commonness among the sampling points.

### 3.4. Comparison with Conventional Field Study

Upon comparing the taxa detected in *C. fluminea* gut contents (20 families, excluding fungi and terrestrial taxa) with the field survey results (123 families) at the family level, 7 families (3 at the genus level and 1 at the species level) were also identified in the field survey (Figure 4). These seven families were Cyclopidae (*Cyclops* sp.), Cyprinidae, Naviculaceae (*Navicula arenaria*), Prorocentraceae (*Prorocentrum* sp.), Selenastraceae, Scenedesmaceae, and Chlorellaceae. Conversely, taxa such as Clausidiidae (*Clausidium* sp.), Paragonimidae (*Paragonimus* sp.), and Strombidiidae (*Strombidium* sp.), which exhibited high FOO and RRA values, were not corroborated by the field survey. In addition, we noted *Oncorhynchus* sp., which was not found in general field surveys. According to the reference report, *Oncorhynchus keta* (Chum Salmon), which was not confirmed in regular surveys, was found during additional monitoring targeting only this species.
Figure 4Venn diagram illustrating the taxa detected at each sampling point and a conventional field survey based on family level. Fungi and terrestrial insect marked with * were not considered in the comparison with the field survey. Salmonidae was placed outside the blue circle (the field survey), as confirmed through a targeted monitoring focused on Chum Salmon (*Oncorhynchus keta*), rather than in a regular field survey.
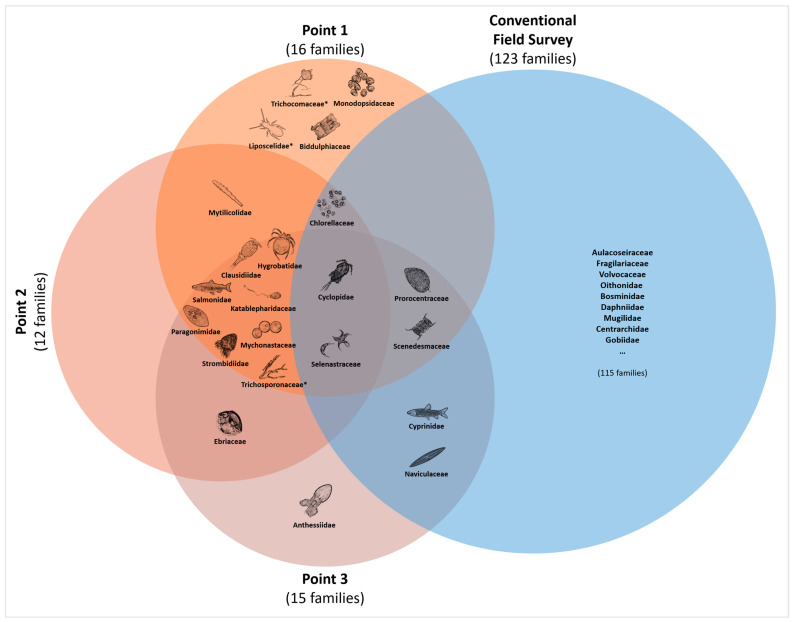



## 4. Discussion

In the present study, we examined the potential use of *C. fluminea* as an eDNA sampler to assess the biodiversity of the Nakdong River Estuary. We extracted eDNA from the gut contents of *C. fluminea* and used the 18S V9 primer to investigate overall biodiversity at the study sites. The metabarcoding results recovered 23 taxa belonging to different classifications, including fish, copepods, diatoms, green algae, which represented a wide range of habitat environments. We also ascertained that out of total 20 families, 7 (35%) were also documented in the conventional field survey and detected elusive migratory fish and planktonic taxa that might be overlooked in the field survey.

### 4.1. Potential of C. fluminea as an eDNA Sampler for Supplementary Biodiversity Monitoring

An ideal eDNA sample for biodiversity monitoring should be capable of detecting a taxonomically wide range of organisms, efficiently providing an accurate reflection of habitat biodiversity [18]. In light of this perspective, our results support the potential of *C. fluminea* to supplement other monitoring methodologies, rather than completely replace them, by providing a perfect illustration of habitat biodiversity.

First, similar to general eDNA samples such as water and soil, eDNA from *C. fluminea* can recover a broad spectrum of taxa, including the species that are challenging to identify through morphological identification alone. These clams have an efficient filtering mechanism [36], indicating their potential as a biological indicator [37,38], and do not represent strong selectivity for food [39], enabling the detection of diverse species. In this study, we successfully identified a wide range of taxa from *C. fluminea* eDNA (Table 2). Notably, although *Clausidium* sp. and *Strombidium* sp. displayed high FOO and RRA values in our results, they were not observed in the field survey. Furthermore, ‘unidentified copepodites and nauplii’ consistently appeared as dominant or sub-dominant species throughout the entire field survey, highlighting difficulties in identifying these species. This represented that eDNA from *C. fluminea* is not limited to specific biological groups, suggesting the presence of unrecognized species, and overcomes and complements the identification limit of the field surveys.

Second, *C. fluminea* eDNA presents an accumulated record within their body over a relatively long period, and present the possibility of detecting rare species that can be overlooked in other monitoring methods. Conventional methods typically yield a ‘snapshot’ of the species present during the survey period, potentially missing species with limited populations (i.e., endangered species) or unique ecological traits (i.e., migratory fish) [7,8]. Water samples can encounter the same issue as the conventional surveys if the volume or duration of filtration is insufficient [40,41]. However, *C. fluminea* rarely leaves its habitat, and provides eDNA accumulated in the body for a longer time. This indicates the possibility of better detection of the missing species despite eDNA degradation within the digestive tract. In this study, we identified elusive migratory fish, *Oncorhynchus* sp., which were not found in conventional field survey. This result suggested that *C. fluminea* eDNA can serve as a puzzle to fill the missing parts of habitat biodiversity.

Third, this approach can be applied in various aquatic ecosystems, easily integrated with other biological monitoring methods. The bivalves are commonly distributed worldwide and exist in various habitats [42,43], so they offer extensive applicability and can be readily collected during field surveys, especially when they are included in the subject of comprehensive monitoring, or when the water is shallow. In other words, additional information can be obtained from the clams “simply picked up during the field sampling”. In particular, when investigating extensive ecosystems with multiple habitat characteristics (e.g., important wetlands or nature reserves), using such a readily accessible natural eDNA sampler may be a way to identify hidden biodiversity and enhance the efficiency of species discovery.

Fourth, eDNA retained within the *C. fluminea* may be less affected by external environmental factors. In this study, we measured and compared water quality parameters at sampling points and confirmed the difference in salinity. Salinity is one of the factors impacting the preservation of eDNA, with higher salinity levels correlating with elevated eDNA concentrations in water [44]. However, statistical analysis of species composition in each point (RRA data) showed no significant difference between the three sampling points (Table A2). Interestingly, even the sampling point with the highest salinity (point 2) showed the least number of OTUs. These results suggest the possibility that eDNA within *C. fluminea* may be less affected by several environmental factors, resulting in better preservation than other environmental samples.

### 4.2. Challenges of C. fluminea as an eDNA Sampler and Future Research

Our sample species, *C. fluminea,* is a typical benthic filter feeder [45], and the possibility of dominant detection of benthic organisms cannot be ruled out. Additionally, in the estuary where freshwater and seawater are mixed, seawater with high density often flows under the freshwater, forming a vertical salinity gradient in the brackish zone [46,47]. Salinity is an important factor that determines the distribution of aquatic organisms [48,49], suggesting that the unique characteristics of this habitat raise concerns about dominant detection of marine species which prefer saltwater. Therefore, when employing bivalves like *C. fluminea* as eDNA samplers, it is imperative to consider the traits of the selected species and the environmental conditions of the research site. Although our study did not demonstrate a pronounced bias toward benthic or marine organisms (Table 2 and Table A1 ), this issue may be a critical consideration when trying to study biodiversity through eDNA of bivalves.

There are few cases of eDNA extraction from bivalves for biodiversity monitoring. This has led to an absence of established protocols for initial sample processing methods, and several studies have shown distinct results depending on their sample treatment method. According to Heo et al. [50], the gut content of Asian clams (*C. fluminea*) recovered relatively fewer OTUs than pseudo-faeces. Jeunen et al. [20] reported that eDNA could not be obtained from the gills of blue mussels (*Mytilus galloprovincialis*). These results suggest that the sample processing method should be carefully selected, and that various methods need to be tried in future studies.

In present study, we assessed the potential of bivalves as an eDNA sampler. However, to fully understand their availability, comparison with water samples will be essential in further studies. Filtered water samples satisfy the conditions of the most ideal eDNA sample [11,18,51]. Therefore, future studies should investigate whether it is worth using *C. fluminea* eDNA compared to water samples under various experimental conditions and environments. In particular, it may be beneficial to explore scenarios wherein the analysis of water samples is considered challenging. For example, research in (1) sites where water flow is irregular, such as closed estuaries, affecting the dispersion and detection efficiency of eDNA [41,52]; and (2) sites where many particles and high turbidity cause filter clogging during water filtration [41,53,54] will provide a new perspective on the value of bivalves as an eDNA sampler.

### 4.3. Limitations in Molecular Analysis

DNA metabarcoding using faeces or gut contents usually poses the risk of excessive detection of self-DNA (i.e., DNA of the sample species itself) or the DNA of the species interacting with the sample species (for example, parasites and symbionts) [55,56,57]. It may be excessively amplified during DNA processing and detected in much larger amounts than prey DNA, leading to non-informative results [56,58]. Our study also identified ten OTUs of *C. fluminea* and one OTU of *Mytilicola* sp., which are known to be intestinal parasites of bivalves. Therefore, considering alternatives, such as blocking oligonucleotides that prevent self-DNA amplification, is necessary to avoid these biases in future studies. 

The 18S rRNA V9 barcode is suitable for DNA metabarcoding for dietary analysis because of its ability to amplify degraded DNA and detect a relatively broad range of eukaryotic organisms [59,60]. However, because the targeting region is relatively short, the 18S rRNA region is considered limited for distinguishing taxonomically close species [61,62]. To compensate for this, we cross-referenced the database and confirmed the indigenous presence of each taxon. Subsequently, species that had never been documented in South Korea were excluded. This process minimised the risk of false-positives caused by short amplicons compared to a simple program that selects taxa with the most similar sequences from the BLASTn results. Nevertheless, in pursuit of finer-grained results at the species level and detection of newly introduced species, future studies must consider a variety of primer combinations, to ideally analyse the decomposed DNA from the sample species.

## 5. Conclusions

In this study, we explored the potential use of *C. fluminea* as an eDNA sampler to supplement the biodiversity monitoring of Nakdong River Estuary. Our analysis of the gut contents of *C. fluminea* has revealed a broad taxonomic spectrum of organisms and provided a chance to uncover hidden biodiversity that may be overlooked in conventional field surveys due to rarity or identification challenges. Additionally, we suggest that the ecological properties of bivalves including *C. fluminea* strongly support their suitability as eDNA samplers for supplementary biodiversity monitoring.

## Figures and Tables

**Figure 1 biology-12-01245-f001:**
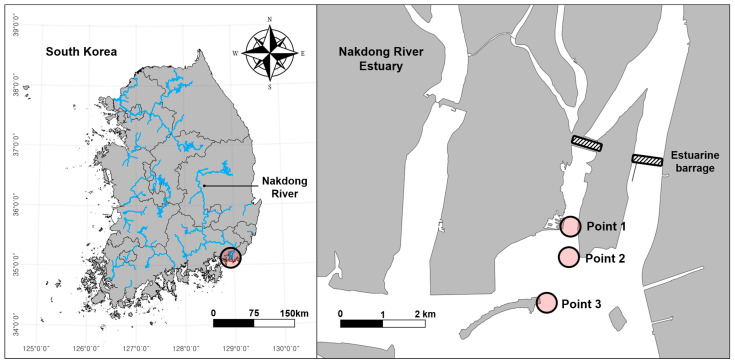
Map of the sampling points in the Nakdong River Estuary.

**Figure 2 biology-12-01245-f002:**
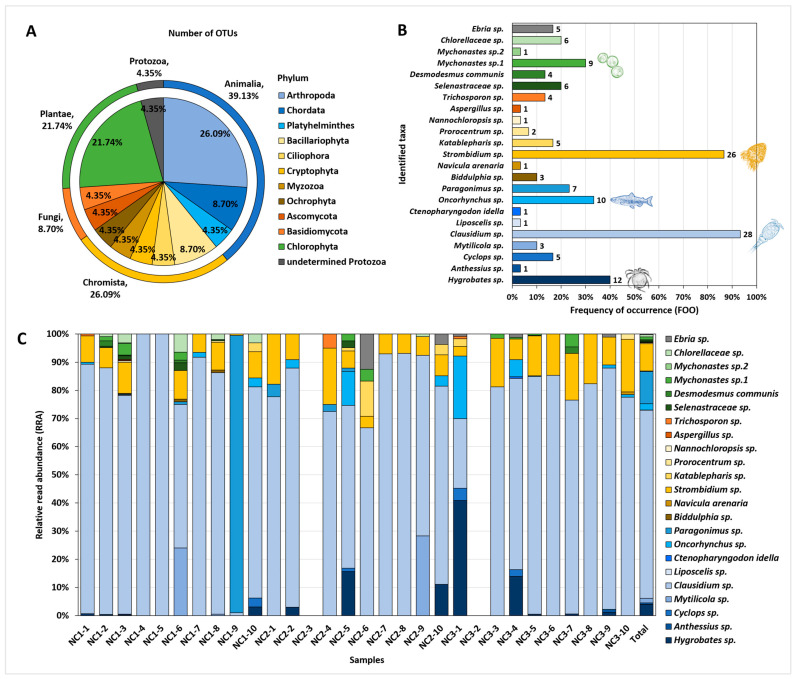
Overall taxa composition detected from the gut contents of *C. fluminea* (*n* = 30). Each colour sector represents a different kingdom group (blue: Animalia, yellow: Chromista, orange: Fungi, green: Plantae, grey: Protozoa). (**A**) The number of OTUs represented as a proportion based on each phylum and kingdom group. (**B**) Frequency of occurrence (FOO) for each OTUs. The number at the edge of the bars refers to the number of *C. fluminea* samples in which the taxon detected. (**C**) Relative read abundance (RRA) for each OTU. The sample name represents each of the 10 samples collected from three different sampling points.

**Figure 3 biology-12-01245-f003:**
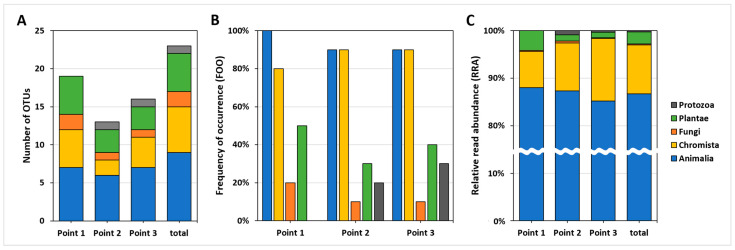
Overall taxa composition (kingdom level) detected from the *C. fluminea* gut contents at each sampling point. (**A**) The number of OTUs at each sampling point based on kingdom group. (**B**) Frequency of occurrence (FOO) at each sampling point based on kingdom group. (**C**) Relative read abundance (RRA) at each sampling point based on kingdom group.

**Table 1 biology-12-01245-t001:** Table of water parameters for each sampling point in the Nakdong River Estuary.

	DO (mg L^−1^)	DO (%)	pH	Temp. (°C)	Conduc. (µS cm^−1^)	Salinity (ppt)	Alkal. (mg L^−1^)	Tur. (NTU)
Point 1	7.00	83.5	7.83	24.1	16,716	10.0	84	3.68
Point 2	6.46	77.5	7.97	24.3	21,666	14.2	90	3.68
Point 3	6.73	80.7	7.99	24.5	20,436	12.3	84	3.97

DO = dissolved oxygen; Temp. = water temperature; Conduc. = electrical conductivity; Alkal. = alkalinity; Tur. = turbidity.

**Table 2 biology-12-01245-t002:** List of identified taxa in the gut contents of *C. fluminea* (*n* = 30) and BLASTn results of each taxon. The OTUs were taxonomically classified according to the WoRMS database.

Kingdom	Phylum	Genus + Species	Max Score	Identity (%)	Query (%)	Genbank Accession	Level
Animalia	Arthropoda	*Hygrobates* sp. ^1^	224	90.06	100	LC552029.1	genus
		*Anthessius* sp.	259	93.18	100	AY627002.1	genus
		*Cyclops* sp.	302	97.73	100	AY626998.1	genus
		*Mytilicola* sp.	309	98.30	100	AY627005.1	genus
		*Clausidium* sp.	270	94.83	100	JF781553.1	genus
		*Liposcelis* sp.	244	96.00	81	AY077779.1	genus
	Chordata	*Ctenopharyngodon idella*	318	98.34	100	XR_007928648.1	species
		*Oncorhynchus* sp.	327	99.44	100	XR_008060685.1	genus
	Platyhelminthes	*Paragonimus* sp.	283	95.03	100	LT855189.1	genus
Chromista	Bacillariophyta	*Biddulphia* sp.	276	96.95	94	JX401228.1	genus
		*Navicula arenaria*	322	100.00	100	KJ961668.1	species
	Ciliophora	*Strombidium* sp.	298	99.39	93	MZ823795.1	genus
	Cryptophyta	*Katablepharis* sp.	324	99.44	100	KJ925151.1	genus
	Myzozoa	*Prorocentrum* sp.	270	94.83	100	MK405477.1	genus
	Ochrophyta	*Nannochloropsis* sp.	305	99.40	100	KU900229.1	genus
Fungi	Ascomycota	*Aspergillus* sp.	320	99.43	100	NG_063229.1	genus
	Basidiomycota	*Trichosporon* sp.	320	99.43	100	MN268783.1	genus
Plantae	Chlorophyta	*Selenastraceae* sp.	309	98.84	100	KT833591.1	family
		*Desmodesmus communis*	316	99.43	100	KF864475.1	species
		*Mychonastes* sp.1	311	98.85	100	OM415709.1	genus
		*Mychonastes* sp.2	316	99.43	100	X73996.1	genus
		*Chlorellaceae* sp.	270	94.29	100	AJ131691.1	family
Protozoa	undetermined	*Ebria* sp.	255	92.74	100	DQ303923.1	genus

^1^ *Hygrobates* sp. was not found in the WoRMS database, so the NCBI taxonomy browser was used instead.

## Data Availability

The datasets generated and/or analyzed during the current study are available in the NCBI repository under accession number SAMN35796656.

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
