# Peer review of "Molecular Diet Analysis of Asian Clams for Supplementary Biodiversity Monitoring: A Case Study of Nakdong River Estuary"

_biology, 2023, doi:10.3390/biology12091245_

Round 1
Reviewer 1 Report (Previous Reviewer 1)
Second review for the paper "Molecular diet analysis of Asian Clam for estuarine biodiversity monitoring: a case study of Nakdong River Estuary" by Kanghui Kim, Gea-Jae Joo, Kwang-Seuk Jeong, Ye-Lim Lee, Donghyun Hong, and Hyunbin Jo submitted to "Biology".
Upon recent perusal of the present paper, I penned down various elements that necessitated improvement.
The authors have responded adequately to my earlier remarks, yet I observed they had incorporated methods unsuitable for carrying out comparisons of the gathered data at disparate junctures. Their claim of employing Analysis of Variance (ANOVA) or Kruskal-Wallis Test (KWT) for observing the "differences in the detected taxa between three sampling points" (L 209-210), leaves one grappling with the puzzle of how univariate methodologies were possibly deployed to discern variations between communities. It is therefore my contention that utilization of a multivariate approach would be more conducive in these circumstances, as seen in Permutational Multivariate Analysis of Variance (PERMANOVA) or Analysis of Similarities (ANOSIM). I therefore recommend that the authors make use of such analytical tools not only to conduct a comparative study of their data but to afford an accurate juxtaposition against field data as well.
Minor
Author Response
I would like to extend my sincere gratitude to you for your thoughtful review of our manuscript. All your feedback, particularly concerning the statistical analysis, has been invaluable in enhancing the quality and clarity of our research.
We have carefully considered your suggestions and have incorporated Permutational Multivariate Analysis of Variation (PERMANOVA) into our data analysis. Consistent with our previous analysis, we have confirmed that there is no significant difference in the detected organisms among the three sampling sites. Detailed descriptions of these findings can be found in the manuscript. Please see lines 194-198 (Materials and Methods section) and 268-271 (Results section).
Additionally, we attempted to apply a multivariate approach to compare our results with those of the field survey. However, the reference report provided integrated information on the survey results, making further analysis impracticable. We apologize for the lack of a more in-depth analysis in this regard and appreciate your thoughtful suggestions.
Your comprehensive and insightful comments have greatly improved our research. Once again, thank you for the time and effort you invested in your assessment.
Reviewer 2 Report (Previous Reviewer 2)
I found this manuscript improved if compared with its first submission, with more accuracy in methods and limitations considerations. I have no other comments in this regard.
Author Response
We are sincerely grateful for your valuable and constructive review. Your feedback has significantly enhanced the quality of our manuscript. We would also like to express our appreciation for your positive comments regarding the improvements made to the text. Once again, thank you for the time and effort you invested in your assessment.
Reviewer 3 Report (Previous Reviewer 3)
N/A
A moderate editing of the English language is required
Author Response
I would like to express my sincere gratitude for your valuable review of our manuscript. We have carefully considered your comments and conducted an overall English correction through the Editage Co. service. We believe that your feedback has significantly improved the quality and clarity of our research. Once again, thank you for dedicating your time and expertise to evaluate our manuscript.
Reviewer 4 Report (Previous Reviewer 4)
I cannot see that this manuscript proves anything. The information about water quality is completely ignored in the Discussion. Why was this information included if it is not worth discussing? Of course the eDNA in the clams includes DNA from some of the organisms in this system. That is a necessary condition for using this approach to assess diversity, but it is far from sufficient. There is a big difference between necessary and sufficient.
Author Response
We sincerely appreciate your thoughtful review of our manuscript. All your comments provided us with an invaluable opportunity to thoroughly revisit our manuscript, and we have made earnest efforts to address the issues raised.
Regarding your comment that our verification of this approach is insufficient, we have rewritten the discussion section to provide stronger support for the idea that C. fluminea eDNA may be a promising eDNA sampler to supplement other monitoring methods. The potential and the evidence supporting it are presented in four main points in Section 4.1. Please refer to Line 312-362.
Furthermore, we deeply agree with your point about the absence of a discussion on water quality data.
We originally presented this data only as baseline information about the survey points' environment. However, we have now compared the water quality data of each point with the metabarcoding results and discussed what this means for the availability of C. fluminea. This content has been included in Section 4.1, which deals with the potential of C. fluminea. Please see Line 353-362.
Your perceptive and insightful comments have greatly enhanced the quality and clarity of our research.
Once again, thank you for dedicating your time and expertise to evaluate our manuscript
Round 2
Reviewer 1 Report (Previous Reviewer 1)
The authors have revised the paper accroding to my comments.
Minor
Reviewer 4 Report (Previous Reviewer 4)
no further comments
This manuscript is a resubmission of an earlier submission. The following is a list of the peer review reports and author responses from that submission.
Round 1
Reviewer 1 Report
Review for the paper "Molecular diet analysis of Asian Clam for estuarine biodiversity monitoring: a case study of Nakdong River Estuary" by Kanghui Kim, Gea-Jae Joo, Kwang-Seuk Jeong, Ye-Lim Lee, Donghyun Hong, and Hyunbin Jo submitted to "Biology".
General comment.
Recent advancements in the acquisition and examination of environmental DNA (eDNA) introduce a complimentary strategy that may serve as an outstanding solution to the persistent problem of gathering comprehensive regional species distribution information; a problem that is typically amplified in remote and otherwise challenging environments due to the constraints of conventional methodologies. The technique of eDNA analysis allows for the discovery of DNA remnants lingering in water deriving from macro-organisms. The process of obtaining water samples for eDNA inspection could lead to faster sample collection, considerably decrease the costs affiliated with data collection and shipping, and present a less destructive method, given there is no necessity for the tampering or expert handling of organisms. Furthermore, the process of eDNA metabarcoding could facilitate the identification of millions of varied DNA fragments per sample, thereby providing an intensely robust procedure to survey aquatic biodiversity. Consistently performed eDNA surveillances can potentially act as a tool to assess enduring alterations in biodiversity, such as the recognition of the loss and decline of native species, the introduction and expansion of non-indigenous species, and variations in the community structure. However, it should be highlighted that the technique of species detection using eDNA can show variation in its effectiveness, fluctuating as a function of the various population densities, life history traits, shedding rates, local environmental conditions, and technical procedures in use, such as the extent of sequencing efforts and issues of primer biases. Additionally, there are significant apprehensions linked with eDNA metabarcoding, including uncertainties over its capacity to accurately identify species by sequences and the relatively unknown ecological dynamics of eDNA in coastal ecosystems. These concerns are of immense importance and require rigorous investigation prior to making any attempts at comparing marine biodiversity across spatial and temporal scales using this particular technique. Up until now, very little is known about the eDNA metabarcoding method in terms of its capability to perform surveys on enduring variations in marine coastal biodiversity. For the first time, the authors have delved deeply into studying the gut content of Corbicula fluminea through eDNA metabarcoding. Their attention was primarily captured by the assessment of the encompassing biodiversity present in the Nakdong River Estuary. The authors identified 21 distinct operational taxonomic units, thereby concluding that this method proves its applicability for monitoring initiatives and biodiversity studies. While the paper is well-articulated and supported with illustrations, it becomes apparent that additional statistical analyses and discussions are necessitated, in order to clarify the authors' claims and adequately answer the question of the capacity of their eDNA survey to improve, or alternatively, not improve biodiversity research.
Recommendations.
Line 66: In their assertion that "The bivalve C. fluminea is widely distributed in the Nakdong River Estuary", the authors should fortify this statement by providing appropriate citations. Additional details about the distribution patterns and abundance/biomass of this mollusk in this area, preferably for each study site, would significantly enrich the discussion. It would also be beneficial to include information about the presence of other bivalve species in this area.
Line 110-111: It is necessary for the authors to indicate the maturity status of the mollusks within the study.
Line 159: Where the authors state that "we excluded taxa that were not present at the study sites", readers would greatly benefit from having details of the datasets used for this exclusion process. The authors should clarify if this exclusion was based on data gathered from the field or if they relied solely on literature sources for this process.
The term used to represent the unit "µL–1" by the authors ("uL") is incorrect. They should revise the use of this unit throughout the text.
Lines 183-184: Here, the authors assert that "The other parameters (that is, dissolved oxygen, pH, water temperature, alkalinity, and turbidity) were not significantly different among the study sites". The words "significant" or "insignificant" differences indicate that a valid statistical analysis was performed, and as a result, the p-value was either lower or higher than what was established as significant (usually 0.05). However, in this situation, there were no apparent statistical analyses conducted to compare water-quality parameters. The authors should either conduct relevant statistical analysis or revise their wording of this sentence to reflect accurately what they found.
Table 2: This table should be updated with the phylum for each taxa for a clearer understanding. The table's title indicates to the readers an expectation of information about the number of mollusks contained within each taxon at each study site. Therefore, the authors should include these data and clarify what the current values presented in the table represent.
Section 3.3: There is ambiguity in the text concerning significant differences among study sites. To clear this uncertainty, the authors should employ a valid statistical analysis.
Section 3.4: For an estimation of the expected number of operational taxonomic units (OTUs) within the study area, the authors should evaluate the Chao2 index.
Line 264: The statement "These results suggest that species and genera are present at the study sites" is redundant considering that they studied a naturally non-sterile system.
Though the authors posited that their results can be utilized for monitoring the aquatic environment, they also pointed out that the methodology employed has some considerable limitations. As such, more detailed discussion on the applicability of this method for biodiversity studies is needed. Additionally, the authors should confront the contrast by discussing and comparing the diversity found in Corbicula fluminea's gut content and in the field. What was new in assessing the overall biodiversity in the Nakdong River Estuary when applying this eDNA screening?
Specific remarks.
L 3. Consider replacing “Nakdong River Estuary” with “Nakdong River Estuary”
L 72. Consider replacing “the potentials and limitations” with “the potential and limitations”
L 83. Consider replacing “within a brackish area” with “within the brackish area”
L 84. Consider replacing “Site 1 is closest from the barrage)” with “Site 1 is closest to the barrage”
L 120. Consider replacing “according to manufacturer’s instructions” with “according to the manufacturer’s instructions”
L 128. Consider replacing “was total 20 uL” with “was 20 µL–1”
L 130. Consider replacing “condition consist of 1 cycle” with “condition consisted of 1 cycle”
L 179. Consider replacing “values ranging” with “values ranging from”
L 244. Consider replacing “The proportion of OTUs in each the study sites” with “The proportion of OTUs at study site”
L 283. Consider replacing “seawater with a high-density flows” with “seawater with high-density flows”
L 305. Consider replacing “for diet analysis because of” with “for diet analysis because of its”
Minor.
Author Response
We sincerely appreciate your valuable and precise comments. Our manuscript has been extensively revised based on all of your feedback. We reanalyzed the raw data from the initial identification process, which led us to uncover several previously overlooked findings. Building upon this enhanced data, we incorporated two additional dietary metrics for a more detailed analysis of the results. We also applied statistical analysis methods to ascertain any differences among sampling points.
Furthermore, in order to thoroughly evaluate the applicability of C. fluminea as an eDNA sampler, we conducted comparative analyses by referencing actual field survey data. The discussion section was reinforced by aggregating all the results. The text was revised overall, so only the amendments to Recommendations and Specific Remarks are marked in RED.
Recommendations.
Line 66: In their assertion that "The bivalve C. fluminea is widely distributed in the Nakdong River Estuary", the authors should fortify this statement by providing appropriate citations. Additional details about the distribution patterns and abundance/biomass of this mollusk in this area, preferably for each study site, would significantly enrich the discussion. It would also be beneficial to include information about the presence of other bivalve species in this area.
- Thank you for your valuable comment. We have been investigating the distribution of fluminea in Nakdong River Estuary, but there was no available data. However, they have been released to the estuary area since 2017 for the purpose of expanding fishery resources, and is widely distributed within the study site. We added the explanation of this to Materials and Methods section. Please see Line 91~93.
Line 110-111: It is necessary for the authors to indicate the maturity status of the mollusks within the study.
- We added the word 'mature' for the explanation of the maturity status. Please see Line 117.
Line 159: Where the authors state that "we excluded taxa that were not present at the study sites", readers would greatly benefit from having details of the datasets used for this exclusion process. The authors should clarify if this exclusion was based on data gathered from the field or if they relied solely on literature sources for this process.
- Thank you for your valuable comments. Accepting the importance of the identification process, we reviewed the overall process again. We have newly referred to the biodiversity database of South Korea and added a detailed description of the process. Please see Line 169-172.
The term used to represent the unit "µL–1" by the authors ("uL") is incorrect. They should revise the use of this unit throughout the text.
- Now revised.
Lines 183-184: Here, the authors assert that "The other parameters (that is, dissolved oxygen, pH, water temperature, alkalinity, and turbidity) were not significantly different among the study sites". The words "significant" or "insignificant" differences indicate that a valid statistical analysis was performed, and as a result, the p-value was either lower or higher than what was established as significant (usually 0.05). However, in this situation, there were no apparent statistical analyses conducted to compare water-quality parameters. The authors should either conduct relevant statistical analysis or revise their wording of this sentence to reflect accurately what they found.
- We revised it by changing the word “significantly” to “considerably”. Please see Line 212.
Table 2: This table should be updated with the phylum for each taxa for a clearer understanding. The table's title indicates to the readers an expectation of information about the number of mollusks contained within each taxon at each study site. Therefore, the authors should include these data and clarify what the current values presented in the table represent.
- We agree with your comments. We revised the table to include only BLASTn results with phylum and kingdom to avoid misunderstanding. Please see Table 2.
- Additionally, instead of removing the number of reads from the table, we replaced it with two additional dietary metrics, FOO (frequency of occurrence) and RRA (relative read abundance), to give a more detailed description of overall taxa composition detected from fluminea gut contents. Please see Line 239~248 and Figure 2.
Section 3.3: There is ambiguity in the text concerning significant differences among study sites. To clear this uncertainty, the authors should employ a valid statistical analysis.
- We performed analysis of variance (ANOVA) and Kruskal–Wallis test to investigate the differences in the detected taxa composition between three sampling points across all taxonomic levels. Please see 267~272 and Figure A2.
Section 3.4: For an estimation of the expected number of operational taxonomic units (OTUs) within the study area, the authors should evaluate the Chao2 index.
- We sincerely appreciate your valuable comments. However, after careful consideration, we have decided to exclude the mentioned results for the sake of a coherent development of the manuscript. While these results supported the adequacy of the experimental methods and sample size, they were deemed incongruent with the overall flow of the paper and of lesser significance. Excluding this data, we focused on understanding the composition of the identified taxa and conducting a comparative analysis with the conventional field survey.
Line 264: The statement "These results suggest that species and genera are present at the study sites" is redundant considering that they studied a naturally non-sterile system.
- We agree with your comments. The sentence has been removed.
Specific remarks.
L 3. Consider replacing “Nakdong River Estuary” with “Nakdong River Estuary”
L 72. Consider replacing “the potentials and limitations” with “the potential and limitations”
- The sentence "Finally, the potentials and limitations of this case study are discussed." has been removed as it was deemed unnecessary for the introduction.
L 83. Consider replacing “within a brackish area” with “within the brackish area”
- Now revised. (Line 89)
L 84. Consider replacing “Site 1 is closest from the barrage)” with “Site 1 is closest to the barrage”
- We rewrote the sentence accordingly, for easy understanding. “Point 1, 2, and 3 were located approximately 2.0 km, 2.7 km, and 3.9 km from the estuarine barrage, respectively.” Please see Line 90-91.
L 120. Consider replacing “according to manufacturer’s instructions” with “according to the manufacturer’s instructions”
- Now revised. (Line 127)
L 128. Consider replacing “was total 20 uL” with “was 20 µL–1”
- Now revised. (Line 136, 144)
L 130. Consider replacing “condition consist of 1 cycle” with “condition consisted of 1 cycle”
- Now revised. (Line 138, 146)
L 179. Consider replacing “values ranging” with “values ranging from”
- Now revised. (Line 213)
L 244. Consider replacing “The proportion of OTUs in each the study sites” with “The proportion of OTUs at study site”
- As the figure was modified, the description was rewritten. Please see Line 274~277 and Figure 3
L 283. Consider replacing “seawater with a high-density flows” with “seawater with high-density flows”
- We rewrote the sentence accordingly for easy understanding. “Additionally, in the estuary where freshwater and seawater are mixed, seawater with high-density often flows under the freshwater, forming a vertical salinity gradient in the brackish zone.” Please see Line 342~344.
L 305. Consider replacing “for diet analysis because of” with “for diet analysis because of its”
- Now revised. (Line 361)
Reviewer 2 Report
Dear Authors,
despite the novelty of this study which could give it high scientific soundness, it is affected by a strong limitation in my opinion.
It's true, as correctly stated in the introduction section that eDNA provide taxonomical and ecological information of better quality compared to classical methods (when analyses are well-conducted), but it's essential to compare the results output from eDNA analyses with the classical methods' information to give a real idea of the communities in a specific site, especially in transitional or extreme environments. Have you carried out investigations in these regards supporting genomic analyses? In the present form, they were totally missed. Is a checklist of this area based on traditional taxonomical methods still existing?
Moreover, the experimental protocol is too essential in my opinion, why the authors have chosen a two-step protocol? Please motivate. Were some blocking primers (self-amplification, human cells, prokaryotes, etc) considered? Results exposure appears very clean, no information about refining was added. What about negative and positive PCR control?
Best regards
The reviewer
Author Response
We sincerely appreciate your precise comments. Based on your feedback, we conducted comparative analyses by referencing actual field survey data to thoroughly evaluate the applicability of C. fluminea as an eDNA sampler (Figure 4). We previously considered various primer combinations, including fish-targeting MiFish primers or blocking primers, but as precedent research, we conducted the experiment simply without a detailed primer design. In the PCR process, both negative and passive controls were employed, and this point was further mentioned in the text (Line 130~131). We are pleased to have received your feedback, and we will certainly address the limitations you pointed out in our future research endeavors.
Reviewer 3 Report
This case study highlights bivalve (Corbicula fluminea) DNA metabarcoding in a closed estuarine system that could support the potential application of bivalves as cDNA samplers in diverse aquatic ecosystems. The paper is important and could be accepted after a minor revision.
The authors mentioned that there are potential applications of bivalves as eDNA samplers in diverse aquatic ecosystems. It would be helpful to understand the readers if they specify what are the potential applications.
The discussion is poorly written. It should be concise with updated references.
Moderate editing of the English language is required.
Author Response
We appreciate the valuable and precise comments. Based on your feedback, we reinforced the discussion by including additional references and improved analysis results. Additional comparative analysis with actual field survey data (Figure 4) can support the possibility of using C. fluminea as a natural tool for collecting eDNA, to identify the overall biodiversity of the habitat. Moreover, we incorporated two additional dietary metrics for a more detailed analysis of the results (Figure 2) and applied statistical analysis methods (Table A2, A3) to ascertain any differences among sampling points.
Reviewer 4 Report
I have reviewed manuscript biology-2488168 by Kim et al. The authors hypothesize (lines 67–68) that eDNA analysis of C. fluminea gut contents could reveal the biodiversity of the Nakdong River Estuary. To test this hypothesis, the authors would have to know what the biodiversity of the Nakdong River Estuary was, and then they would have to determine whether eDNA analysis of C. fluminea gut contents revealed that biodiversity. Because the authors have no knowledge of the true biodiversity of the Nakdong River Estuary, and hence there was no way for them to test their hypothesis. The conclusions of the manuscript say, “These findings indicate the potential use of C. fluminea in eDNA metabarcoding for biodiversity investigations.” The study does show that it is possible to do eDNA metabarcoding on the gut contents of C. fluminea, but I am afraid that is about all this study shows. The Conclusions section suggests that the authors would like to carry out biodiversity investigations using eDNA. There are surely ways to carry out biodiversity investigations using eDNA other than sampling the gut contents of clams. An informative study might be a comparison of estimates of biodiversity based on the gut content of clams as well as other methods to see which method or combination of methods is most promising. I just do not see that this study proves anything other than the fact that it is possible to carry out analyses of the eDNA in the guts of clams.
Author Response
We sincerely appreciate your valuable comments. Based on your feedback, we conducted comparative analyses by referencing actual field survey data to thoroughly evaluate the applicability of C. fluminea as an eDNA sampler. These results are presented in figure 4, and the discussion section deals with the detection of rare or planktonic species that are difficult to identify in field surveys. We also addressed some of the value and potential of C. fluminea compared to other eDNA sources. Please check the section 4.1.